# The ART of Link Prediction with KGEs

**Yannick Brunink**                                       Y.BRUNINK@VU.NL
**Michael Cochez**                                        M.COCHEZ@VU.NL
**Jacopo Urbani**                                          J.URBANI@VU.NL
*Vrije Universiteit, Amsterdam*

**Editors:** Leilani H. Gilpin, Eleonora Giunchiglia, Pascal Hitzler, and Emile van Krieken

## Abstract

Link Prediction (LP) in Knowledge Graphs (KGs) is typically framed as ranking candidate entities for a query of the form $(entity, relation, ?)$, with models evaluated on their ability to rank the correct entities for each query. At the same time, Knowledge Graph Embedding (KGE) models used for this task produce unnormalised scores, making it unclear how to interpret their belief in the truthfulness of triples across different queries. Together, these two factors create a blind spot: models can achieve perfect rankings while assigning scores that are not comparable across queries, limiting their utility in downstream tasks or even in identifying the most plausible triples overall. Indeed, this issue becomes clear when test triples are ranked globally and evaluated with IR metrics, revealing that models with unnormalized scores often perform poorly due to inconsistent scoring across queries. To address this problem, we propose a new KGE model, called `ART`, which exploits probabilistic Auto-Regressive modelling and hence is normalised by design. Despite its conceptual simplicity, we show that `ART` outperforms prior art for discriminative and generative LP as well as other post-hoc calibration techniques.

## 1. Introduction

Knowledge Graphs (KGs) are structured representations of knowledge, organised as graphs that model relationships between entities. KGs are widely studied in academia and extensively utilised in industry, driving advancements in research and practical applications (Wang et al., 2017b; Hogan et al., 2021; Zhang et al., 2023). However, a key limitation of KGs is their inherent incompleteness—many links (triples) are missing, reducing their overall utility in real-world scenarios. An important step toward addressing this limitation is Link Prediction (LP), where the goal is to predict which unknown links in a KG are likely to be true, given the observed graph (Lü and Zhou, 2011).

A mainstream approach to solving the LP problem involves Knowledge Graph Embedding (KGE) models. These models transform the entities and relations into continuous vector spaces, to assign each triple (subject, relation, object) a real-valued score through a scoring function (Wang et al., 2017a). To answer queries of the form (subject, relation, ?) or (?, relation, object), KGE models return a ranked list of candidate entities and performance is evaluated by averaging the quality of these rankings using metrics such as Mean Reciprocal Rank (MRR) or hits@K.

While KGE models have shown strong performance on these metrics, the scores they assign to triples—used to produce the rankings—are not normalised (Loconte et al., 2023), which limits the interpretability of the model's beliefs regarding the plausibility of individual triples (Friedman and den Broeck, 2020). For example, when KGE models are used

to predict drug side effects (Yao et al., 2022), the scores of candidates corresponding to the query (paracetamol, hasSideEffect, ?) cannot be compared with those from the query (morphine, hasSideEffect, ?) as they might be on a different scale. In general, empirical evidence of this problem can be obtained by collecting all the scores assigned to candidates of queries, construct a single unified ranking, and assessing its quality using standard Information Retrieval metrics, such as max-$F_1$. Notice that this evaluation methodology is different than Triple Classification (TC) (Socher et al., 2013), which reframes the problem as a binary classification task rather than ranking. While TC also often implies scores to be globally comparable, as it frequently relies on decision thresholds shared across all queries, TC brings the additional challenge of finding a good strategy of making the binary decision.

Although this problem has received some attention, it remains very challenging. In general, existing solutions for producing comparable scores fall into two categories: post-processing methods, where scores are rescaled through a specific procedure, e.g. calibration (Tabacof and Costabello, 2020), and probabilistic frameworks, which aim to produce intrinsically normalised scores. To our knowledge, the only approach in the latter category extends TF models by applying a single non-linear transformation to ensure positive scores, and calculating the normalisation constant (Loconte et al., 2023). Although this results in probabilistic outputs, a single non-linear transformation limits the model's ability to capture complex dependencies in the KG.

**Contributions**  Our first contribution is to establish a principled methodology to evaluate normalisation using a global ranking and IR metrics. The second is a new addition to the portfolio of KGE methods, which consists of a simple, yet powerful, generative KGE model called `ART`. In contrast to other solutions, `ART` is fully auto-regressive generative; hence is normalised by design. Experiments, using our proposed methodology and pre-existing metrics for LP, yield several key insights. First, an evaluation based only on query-based ranking, with metrics such as MRR, does not give a full picture as it does not take into account normalisation. Indeed, unnormalised methods that are leading under MRR do not perform as well as normalised methods when considering our proposed methodology. Among the normalised methods, `ART` achieves the best results on two out of three benchmarks and delivers competitive performance on the third. Finally, our results indicate that intrinsically normalised methods consistently outperform post-processing techniques like calibration.

## 2. Background

**Knowledge Graphs**  A KG is a structured graph-like representation of entities and their relationships. It can be represented as a set of triples of the form $(s, r, o)$ where $s, o$ belongs to a set $\mathcal{E}$ of entities and $r$ from the set $\mathcal{R}$ of relations. KGE models use latent representations of KGs, trained for tasks such as LP. KGE models typically consist of two components. One component represents entities and relations as continuous vectors (embeddings) in a low-dimensional space $\mathbb{R}^d$. These embeddings are designed to capture the semantic meanings and structural information of the entities and relations while preserving the inherent structure of a KG. The second component is a scoring function $\phi : \mathbb{R}^d \times \mathbb{R}^d \times \mathbb{R}^d \to \mathbb{R}$ that assigns a score to the vector representation of every triple, $\phi(\mathbf{e_s}, \mathbf{r_r}, \mathbf{e_o})$. That score is usually a real value in $\mathbb{R}$ and represents the model's belief in the truthfulness of the triple.

Popular KGE models, such as ComplEx (Trouillon et al., 2016), can be interpreted as binary classifiers (Loconte et al., 2023) where each triple corresponds to a distinct Bernoulli random variable, such that $p(Y_{sro}|S, R, O)$. An important limitation is that the scores are unnormalised and that the model's belief in a triple's truthfulness is inherently context-dependent, that is, the scores are only meaningful within the scope of a specific query (Friedman and den Broeck, 2020). Moreover, for binary classification, negative examples are required for training models to distinguish between true and false candidates. KGs typically do not contain negative examples (Tabacof and Costabello, 2020). To counter this, negative examples can be artificially created by sampling unknown triples and assuming that they are false via a technique known as negative sampling (Bordes et al., 2013).

**Generative Models** It may be desirable to learn a KG model that is intrinsically normalised, which means that the scores are probabilities, and that the model describes the processes that produce data (Papamakarios et al., 2021). This naturally motivates a generative approach, which aims to model the joint distribution, $p(\mathbf{x})$, of a dataset $\mathcal{D}$ of $n$-dimensional datapoints $\mathbf{x}$ (Ruthotto and Haber, 2021). The goal is to optimise the closeness between the data and the model distribution. This is achieved by selecting the model parameters $\theta \in \mathcal{M}$ that maximise the log-likelihood of the data $\mathcal{D}$. This learning objective is called Maximum Likelihood Estimation (MLE).

$$\max_{\theta \in \mathcal{M}} \frac{1}{|D|} \sum_{\mathbf{x} \in \mathcal{D}} \log p_\theta(\mathbf{x}) = \mathcal{L}(\theta|\mathcal{D}). \tag{1}$$

In order to learn a valid joint distribution, the total probability mass assigned to all possible datapoints should sum to one. Different classes of generative models have different ways of achieving this. We briefly discuss the ones that are most relevant for this work.

**AutoRegressive Models** Deep AutoRegressive models (ARMs) (Gregor et al., 2014) factorise the joint distribution according to the chain rule of probability, into a marginal distribution and a sequence of conditional distributions. A key simplifying assumption is that each factorised distribution can be modelled as a (generalised) Bernoulli random variable. Because a Bernoulli distribution is completely specified by its mean (the probability of success), this simplifies normalisation: once the mean is predicted, the distribution is fully defined and normalised. Subsequently, if the factorised distributions are normalised locally, the joint distribution is also normalised.

**Energy-based Models** Energy-based models (EBMs) (LeCun and Huang, 2005) can learn arbitrary probability distributions by assigning an energy value to each configuration of variables. EBMs are based on the Boltzmann distribution: $p(x) = \frac{1}{Z}e^{-E(x)}$, where $E(x)$ is the energy function and $Z = \sum_x e^{-E(x)}$ is the partition function. While this formulation is very flexible, computing $Z$ is often intractable as it requires summing all possible configurations. If we apply EBMs to implement KGEs, the normalisation constant, $Z = \sum_{(s,r,o) \in \mathcal{T}} \phi(\mathbf{e_s}, \mathbf{r_r}, \mathbf{e_o})$, requires computing $\mathcal{E} \times \mathcal{R} \times \mathcal{E}$ triples at every training step, which even for small KGs this is infeasible (Loconte et al., 2023). Moreover, non-negative scores have to be enforced. Next, we briefly recap how this has been addressed in the recent work by Loconte et al. (2023).

**KGEs as Probabilistic Circuits (Loconte et al., 2023)** Some of the most popular KGE models (Trouillon et al., 2016; Lacroix et al., 2018; Nickel et al., 2011) are Tensor Factorisation models (Hitchcock, 1927). They are called Tensor Factorisation (TF) models because the triple score is a product of the subject, relation, and object factors (embeddings). For example, the scoring function of CP (Lacroix et al., 2018) is the dot product. These models can be interpreted as unnormalised EBM models (Minervini et al., 2016). However, by interpreting the TF models as a computational graph, they can be cast into a Probabilistic Circuit (PC) (Choi et al., 2020). If a PC is smooth and decomposable, the partition function can be calculated in $O((|\mathcal{E}| + |\mathcal{R}|) \cdot cost(\phi))$ time, enabling efficient normalisation (Choi et al., 2020). Subsequently, non-negativity is enforced by extending the models with a non-linear operation, e.g., the square operator (Loconte et al., 2023).

## 3. Our Approach

A natural starting point for learning a generative KGE model is to normalise the distribution of traditional KGE models $p(Y_{sro}|S, R, O)$ by learning the joint distribution $p(Y, S, R, O)$. This yields a normalised model that can assign probabilities to triples (Lasserre et al., 2006). However, since $Y$ is not observed and must be approximated via negative sampling, its generative interpretation is unclear. By directly modelling $p(S, R, O)$, we no longer need to rely on artificially constructed negatives.

The prior work by Loconte et al. normalises this distribution, extending TF models with a single non-linear transformation to ensure non-negative scores. A property of TF models, multi-linearity (Kolda and Bader, 2009), allows each embedding dimension to contribute independently to the final score, resulting in strong classification performance. However, when such KGE models are extended with non-linear operations—such as squaring—they lose their core characteristic of multilinearity and can no longer be considered TF models. This change is not only structural but also detrimental: they tend to perform significantly worse on the original ranking task (Lacroix et al., 2018). Moreover, moving from binary classification to modelling a probability distribution results in a shift in objective. Our proposal below aims at addressing these limitations, with a different approach.

### 3.1. ART

In many domains, (deep) ARMs are used to model high-dimensional data by stacking a large number of simplified distributions—typically low-capacity conditionals like Bernoulli or categorical distributions. Although effective, this often reflects a trade-off between expressivity and tractability (Salimans et al., 2017; van den Oord et al., 2016) because the large number of factorized distributions acts as an approximation. In contrast, in our case, there are only three variables that we can use for factorizing the joint distribution: subject, relation, and object. Since they are discrete and finite, the autoregressive factorisation is not a simplifying approximation, as it is when ARMs are used in other domains, but a natural decomposition, allowing us to retain both tractability and expressivity.

Therefore, we operationalise this decomposition in our proposed model `ART` using the chain rule of probabilities as follows:

$$p(S, R, O) = p(S) \cdot p(R|S) \cdot p(O|R, S) \tag{2}$$

While this decomposition is exact, in general it still requires summing over all possible $(S, R, O)$ configurations to ensure normalisation. Fortunately, the variables $S$, $R$, and $O$ are categorical in our domain since they take on one of a fixed set of possible values (e.g., entities or relations). Hence, each factorized conditional distribution can be implemented using a Softmax. Because each part of the factorised distribution is normalised this way, the full joint distribution is also normalised.

In the remaining, we motivate why we chose this factorization and not other ones.

**Model's size**  KGs typically contain far fewer relations than entities, meaning that the training data offer more coverage of the relations. As a result, the marginal distribution of the relations that we can compute from the frequencies of the training dataset typically tends to better reflect the real one than it does, for instance, for subjects. Therefore, one could wonder why not using an alternative factorisation such as $p(R) \cdot p(S|R) \cdot p(O|R, S)$. The problem is that such formulation would significantly increase the size and computation of the models of the other two factors, since it would require a softmax over all subjects and objects. This is why we decided to implement the factorization in Eq. 2.

**Head vs. tail queries**  Our model defines a joint distribution over triples, $p(S, R, O)$, allowing us to assign a probability to any triple. To efficiently evaluate all candidates for a tail query $(s_1, r_1, ?)$, we compute $p(S = s_1)$ and $p(R = r_1 \mid S = s_1)$ once and multiply them with $p(O \mid R = r_1, S = s_1)$ in a single pass—mirroring the efficiency of standard KGE models. However, this efficient computation only holds in the forward direction (tail prediction). Tasks like Link Prediction and Complex Query Answering (CQA) often require scoring in both directions i.e. head and tail queries. To support head queries, one may decide to implement the factorization $p(O) \cdot p(O|R) \cdot p(S|R, O)$. However, this is not needed, since a known solution for this problem is simply to add inverse triples (Dettmers et al., 2018). This effectively lets the model answer head queries as tail queries on the augmented graph, introducing directionality at the cost of doubling the relation types — but importantly, still using a single model with shared parameters rather than training two separate models.

## 3.2. Architecture

We provide a high-level overview of the core design choices and model structure. We refer to Appendix B.1 for a detailed specification of our architecture.

**Conditional Distributions**  Our conditional distributions—$p(R \mid S)$ and $p(O \mid R, S)$—can be parameterised by any neural network. For example, ConvE (Dettmers et al., 2018), a Deep KGE model, can be interpreted as modelling $p(O_{sr} \mid S = s, R = r)$; it takes embeddings of $(s, r)$ as input and outputs a score for each candidate object. We follow this idea but replace the Sigmoid activation with a Softmax, yielding a proper categorical distribution over all objects. To complete the factorisation, we also add a conditional distribution $p(R \mid S)$, modelled by a network that takes only the subject embedding as input and outputs a distribution over relations. To model both conditionals efficiently, we use an AutoRegressive Transformer (Vaswani et al., 2017), which can handle both timesteps jointly within a single architecture. This is why we call our method `ART` (AutoRegressive Transformer).

**Prior**  A natural choice to implement the marginal distribution $p(S)$ would be to use the raw (relative) train frequency of the subjects. However, this frequency might not accu-

rately represent the "real" distribution. To *learn* the most likely parameters (MLE) for the marginal distribution as well as the other two conditional probabilities, we use a learnable parameter for every logit, and add a learnable Temperature parameter $T$, which allows us to smooth the train frequencies if needed and jointly optimise the objective.

More formally, our models are learned by optimising the exact Maximum Likelihood Estimation (MLE) as follows, where $\mathcal{D}$ is the training set and $\theta$ are the parameters of the model.

$$\max_{\theta} \mathcal{L}(\theta) = \frac{1}{|\mathcal{D}|} \sum_{(s,r,o) \in \mathcal{D}} [\log p_{\theta}(s) + \log p_{\theta}(r|s) + \log p_{\theta}(o|r,s)] \tag{3}$$

## 4. Related Work

We extensively discussed the most related work (Loconte et al., 2023) in Section 3. Other relevant approaches to this problem take different perspectives. AutoRegressive (AR) models have been extensively studied for triple likelihood estimation in Knowledge Graphs (Chen et al., 2021; Yao et al., 2019; Tresp et al., 2021; You et al., 2018). However, these approaches either rely on Large Language Models, focus on local subgraphs, or do not explicitly model the KG data as three Categorical variables. Consequently, they are not amenable for exact Maximum Likelihood Estimation (MLE) as discussed by Loconte et al. (2023). In this work, we focus on learning an exact joint distribution over three *Categorical* random variables using MLE. Hence, we do not consider methods that approximate the joint distribution (Simonovsky and Komodakis, 2018).

The first work that gave a probabilistic interpretation to tensor factorisation KGE models was tractOR (Friedman and den Broeck, 2020). They assume conditional independence betweeen the random variables (factors), such that $p(Y_{sro} = 1|s, r, o) = p(E_s = 1|s) \cdot p(T_r = 1|r) \cdot p(E_o = 1|o)$. Consequently, the number of to be learned independent variables is reduced from $|\mathcal{E}| \cdot |\mathcal{R}| \cdot |\mathcal{E}|$ to $|\mathcal{E}| + |\mathcal{R}|$. Moreover, the decomposition in unary statements enables fast inference in Probabilistic Databases (PDBs) (Suciu et al., 2011). However, the scores are unnormalised, and the conditional independence assumption is problematic, as it rules out any interaction between the subject, relation, and object — which are typically highly dependent in real-world KGs.

Our evaluation protocol relates to Triple Classification (Socher et al., 2013), which classifies triples as true or false using a threshold, implicitly adopting the Closed World Assumption (CWA)—treating all unknown triples as false. Finding the optimal threshold requires calibration (Tabacof and Costabello, 2020) and forces binary decisions. In contrast, we use a ranking formulation aligned with the Open World Assumption (OWA), focusing on whether known triples rank higher than unknown ones, similar to LP, without requiring calibration or binary classification.

## 5. Experiments

**Experimental Setup** We conducted all experiments [1] on an Nvidia RTX A4000 GPU. Deep KGE models tend to have many parameters. To show that our improved expressiveness does not come from increasing model size, we keep our total parameter count relatively

---

1. The code is available at https://github.com/yaaani85/art_kge

Table 1: Statistics of datasets used for the evaluation, including number of entities ($|\mathcal{E}|$), relations ($|\mathcal{R}|$), and data splits. Final columns indicate the number of forward and inverse queries derived from the test set.

| Dataset | $|\mathcal{E}|$ | $|\mathcal{R}|$ | Train | Valid | Test | IR Test (Fwd/Inv) |
|---|---|---|---|---|---|---|
| FB15k-237 | 14,541 | 237 | 272,115 | 17,535 | 20,466 | 10,270 / 10,196 |
| WN18-RR | 40,943 | 11 | 86,838 | 3,034 | 3,134 | 2,694 / 440 |
| OGBL-BIO | 93,773 | 51 | 4M | 163K | 163K | 88,808 / 74,062 |

low: We fix the embedding rank to 150 and adopt a simple decoder-only Transformer with up to four layers, a single attention head, and no positional encoding. For a detailed description, see Appendix B.2.

**Datasets**   We evaluated `ART` and its competitors on datasets which are frequently used in the literature: `FB15k-237` (Toutanova and Chen, 2015), `WN18RR` (Dettmers et al., 2018) and `ogbl-biokg` (Hu et al., 2020). Table 1 reports statistical details for each of them. We use the transductive splits, provided in the original papers, where all entities in the validation and test sets are observed during training.

**Baselines**   We chose to compare `ART` against the following approaches:

- `ComplEx`$^2$ (Loconte et al., 2023): A normalised version of ComplEx that learns exact joint distribution via global normalisation, introduced in Section 3.

- `NBF` (Zhu et al., 2021): Neural Bellman-Ford Networks, which is, to the best of our knowledge, the KGE with the best results on LP.

- `ComplEx` (Trouillon et al., 2016), which is another KGE model that is frequently used in the literature, which leverages complex embeddings and produces unnormalised scores.

- `ComplEx/Cal` (Tabacof and Costabello, 2020): ComplEx with Platt scaling applied post-hoc to calibrate the scores. This was selected as the best method by prior work on calibration.

Our primary comparison is against `ComplEx`$^2$, since both `ART` and `ComplEx`$^2$ return probability distributions over triples. We select `ComplEx` and `NBF` as representative of the leading unnormalised KGE methods. Finally, we include `ComplEx/Cal` as representative for post-hoc calibration.

For `ComplEx`$^2$ and `NBF`, we adopt the configuration provided in the original implementation. To isolate the effects of normalisation and post-processing, we use identical model configurations across `ComplEx`$^2$, `ComplEx`, and `ComplEx/Cal`. This is to ensure that differences in performance are solely due to normalisation or calibration.

**Testset construction**   We construct our candidate set directly from the LP benchmarks, using the triples that are candidates in head and tail prediction queries. This ensures our evaluation is a natural extension of LP rather than a new task — we unify what is typically evaluated per query into a single, global ranking over all candidate triples.

While we could restrict evaluation to a single prediction direction (e.g., tail prediction only, as common in Triple Classification (Socher et al., 2013)), this would test only half of the model's scoring behaviour. Many downstream applications, such as Complex Query Answering (CQA) (Ren et al., 2024), require combining scores from both directions.

However, to prevent duplicates from appearing — for example, when the same triple occurs in both head and tail prediction queries — we include each triple only once, based on its first occurrence. This avoids artificially inflating ranking-based metrics by giving the model multiple opportunities to score the same fact. It also keeps the candidate set from growing unnecessarily large and results in a more challenging and informative evaluation, as the unified ranking must reflect a coherent ordering over head and tail predictions drawn from across the entire benchmark. More details are available in Appendix A.1.

**Evaluation Protocol**   We follow the standard protocol for traditional LP and evaluate on two prediction tasks: object prediction $(s, r, ?)$ and subject prediction $(?, r, o)$. For each test triple $(s, r, o)$, we generate candidates by replacing either the object or subject with all possible entities, and rank these candidates using the model's scores. We use the filtered setting  (Bordes et al., 2013) for the query-based evaluation, and report the mean reciprocal rank (MRR) (Lü and Zhou, 2011). For completeness, we also report Hits@k (Lü and Zhou, 2011) in the Appendix  C.1.

**New proposed metric**   To evaluate normalisation, we propose a new metric in which we construct a single global ranking across all queries rather than separate rankings per query. Instead of evaluating performance on a specific downstream task, we introduce this metric to enable intrinsic evaluation, isolating the normalisation capability without interference from task-specific factors. We argue that this metric serves as a strong proxy for many downstream tasks, as it directly assesses the comparability of scores across different queries.

To this end, the metrics used for query-based ranking are unsuitable because now there is only one ranking. Therefore, we instead rely on standard IR metrics such as Precision, Recall, and $F_1$. Crucially, this evaluation remains fully aligned with the LP setting in every other respect: we use the same filtered candidates, evaluate the same triples, and do not modify the task from ranking to classification.

We report the max-$F_1$ score over all possible thresholds used to compute precision and recall, capturing the model's best precision–recall balance without relying on a learned threshold. This aligns with the Open World Assumption by offering a nuanced evaluation across all triples. Unlike Triple Classification, which requires tuning thresholds on validation data, our approach evaluates performance across all possible thresholds. For completeness, we also provide precision–recall curves in the Appendix C.2 to illustrate this range.

**Model Selection**   After training, we select the best model based on MRR validation using a limited hyperparameter search (see Appendix B.2). While we report the test set max $F_1$ for reference, the threshold is computed post hoc and is not used for model selection. This ensures fair comparison and adheres to standard evaluation practice.

Table 2: Performance comparison using query-based MRR and max-$F_1$ global rankings. For both metrics, higher is better. For max-$F_1$, results in green are either the best ones or very close to the best (max 5% difference) while the ones in orange (red) are the ones that are (much) worse than the best.

| Model | | FB15K-237 | | WN18RR | | OGBL-BIOKG | |
|---|---|---|---|---|---|---|---|
| | | MRR | max-$F_1$ | MRR | max-$F_1$ | MRR | max-$F_1$ |
| ART (OURS) | † | .342 | .273 | .451 | .525 | .832 | .412 |
| COMPLEX$^2$ | † | .300 | .191 | .391 | .074 | .840 | .438 |
| NBF | ◇ | .415 | .181 | .515 | .540 | .811 | .363 |
| COMPLEX | ◇ | .336 | .049 | .470 | .503 | .828 | .278 |
| COMPLEX/CAL | * | .336 | .049 | .470 | .503 | .828 | .278 |

† NORMALISED (GENERATIVE) MODELS, ◇ DISCRIMINATIVE MODELS, * POST-PROCESSED VARIANT OF COMPLEX WITH PLATT SCALING.

## 5.1. Results

Table 2 shows a performance comparison with the MRR computed with query-based ranking and with our proposed metric for measuring normalisation. We make *three key observations.*

**Observation 1: MRR vs max-$F_1$**   First, we note that LP metrics (query-based MRR), which were extensively used in the literature, are not a reliable indicator of performance in tasks that require normalisation. This can be deduced from the fact that unnormalized methods, like NBF, which returns the best scores on MRR, do not perform equally well with max-$F_1$. Second, we also observe that methods that are intrinsically normalised (ours and ComplEx$^2$) outperform the other competitors with our new proposed metric (max-$F_1$) on two of the three datasets. On the third one, WN18RR, our method returns a score that is very similar to NBF. The reason is that it is well-known there is a distribution shift between the train and test triples in WN18RR (Loconte et al., 2023). This is known as the Domain Adaptation problem (David et al., 2010). For instance, 19% of the subjects in the test set are not in the train set, and 95% of subject-relation pairs in the test set are not in the train set. This means that the testset is not a faithful representation of the distribution that we can learn from the trainset. Notice that despite this problem, our method is still able to predict as accurately as the state-of-the-art.

**Observation 2: ART vs ComplEx$^2$**   Contrasting with ComplEx$^2$, which relies on a single non-linear operation, our model employs multiple non-linearities encoded in the Transformer, providing greater expressiveness. While using only half of the parameters, see Appendix C.3, ART had superior performance on both MRR and max-$F_1$, making our approach the preferred choice when comparable scores are required. Moreover, because ART factorizes the joint distribution as $p(S) \cdot p(R|S) \cdot p(O|R, S)$, it can better adapt to distribution shifts, like those in WN18RR, than ComplEx$^2$.

**Observation 3: Intrinsic Normalisation vs. Calibration**   Post-processing methods such as calibration or score normalisation applied after training do not affect MRR (Tabacof

and Costabello, 2020), and similarly have no impact on our IR metrics. While calibration shifts the optimal max-$F_1$ threshold from its original value (e.g., 70) to 0.5 after calibration, as shown in Appendix C.2 — it does not improve the max-$F_1$ score. This demonstrates that calibration changes the presentation of confidence scores but does not alter the model's underlying belief distribution. Therefore, this suggests that encoding normalisation inside the model is a more effective approach than post-hoc calibration.

**Uncertainty Quantification**   While ART improves performance on IR metrics compared to unnormalised approaches, it also provides a measure of likelihood. In this context, a common metric for evaluating uncertainty is Expected Calibration Error (ECE) (Tabacof and Costabello, 2020). Previous work has shown that normalisation improves ECE scores (Loconte et al., 2023). However, while this distinction is clear when comparing unnormalised and normalised models, using ECE to determine which of two normalised models is better is ambiguous due to the OWA inherent in KGs. In Appendix C.3, we discuss this issue considering a scenario in which all unknown triples with high probability are assumed true rather than false, resulting in markedly different outcomes.

## 6. Conclusion and Future Work

**Summary**   This paper highlights a critical limitation of many state-of-the-art KGE models. The prediction scores of a KGE model for multiple queries are not directly comparable, which significantly reduces their applicability in downstream tasks that, for example, require score aggregation across queries.

The standard evaluation protocol, which focuses on query-based ranking, overlooks this issue. To fill this gap, we propose another evaluation methodology that exploits global ranking and IR metrics as proxies for assessing effectiveness in scenarios where scores from different queries must be combined. In addition, we introduce a novel generative KGE model, ART, that produces intrinsically normalised scores. Our results on multiple benchmarks demonstrate that ART is substantially more effective at generating scores that are consistent and comparable across queries.

**Future Work**   Normalisation is useful in many downstream tasks. One is Complex Query Answering (CQA). Recent work shows that CQA performance, when using the same unnormalized models evaluated in this study, is notably degraded with very complex queries (Gregucci et al., 2025). This motivates future work to apply ART to CQA. This requires a dedicated study since answering complex queries efficiently requires additional tasks, like query planning or cardinality estimation, which is challenging if the underlying KB is provided by a KGE. Another interesting avenue for future work is KB Completion, which calls for a system that is capable of making binary decisions (Triple Classification) regarding the truthfulness of facts. Next to the challenge of identifying suitable strategies for deciding when a triple is true, it is worth further studying if the probabilities are overconfident with specific classes of entities, or whether there is any other source of bias that stems from training in a generative setting.

In conclusion, this work advances the development of KGE models with reliable score normalisation, enabling more robust evaluation, improved comparability between queries, and enhanced applicability in a wider range of downstream settings.

## Acknowledgments

We thank Ruud van Bakel, Emile van Krieken, Jakub Tomczak, Wilker Aziz, and especially Philip Boeken for helpful discussions. We also thank the DAS-6 project (Bal et al., 2016) for providing access to the computing infrastructure used in this work. Michael Cochez is partially funded by the Elsevier Discovery Lab, partially funded by the Graph-Massivizer project, funded by the Horizon Europe programme of the European Union (grant 101093202), and supported by a gift from Accenture LLP. His work on this publication is in part based upon work from COST Action CA23147 GOBLIN - Global Network on Large-Scale, Cross-domain and Multilingual Open Knowledge Graphs, supported by COST (European Cooperation in Science and Technology, https://www.cost.eu).

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

## Appendix A. Experimental Setting

### A.1. Information Retrieval Candidates

As detailed in the main paper, our Information Retrieval (IR) benchmark is derived from the LP test sets. We construct the IR set by first executing forward (tail) queries of the form $(s, r, ?)$, and storing these triples. We then traverse the test set again, executing corresponding inverse (head) queries $(?, r, o)$. If a triple has already been included as a forward query, it is skipped for inverse inclusion. This ensures that each test triple appears at most once in the IR benchmark, avoiding duplication and promoting diversity in query direction.

This method typically leads to a roughly balanced distribution between forward and inverse queries, as seen in `FB15k-237` and `OGBL-BIO`. However, `WN18RR` shows a strong imbalance favoring forward queries. This is not a consequence of our construction method, but a result of the dataset's inherent structure—many `WN18RR` test triples have tail entities that appear far more frequently or earlier than the corresponding head counterparts, causing them to be selected first during traversal. As such, most inverse candidates are filtered out, leading to the observed skew.

While we could have artificially rebalanced the IR set—for example, by resampling or enforcing symmetry—we chose not to. Instead, we prioritise a reproducible and transparent generation process. Selecting the first occurrence of a triple, regardless of direction, is deterministic and straightforward. This avoids arbitrary rebalancing heuristics and makes the construction method easy to implement on other datasets. Please see the last column of Table 1 for the resulting number of forward and inverse triples included in our evaluation.

The code used to generate the IR benchmark is included in the supplementary material for full reproducibility.

## Appendix B. Details of `ART`

### B.1. Architecture

**Attention is all you need.**  Our autoregressive model `ART` is based on a standard Transformer (Vaswani et al., 2017) architecture composed of stacked self-attention blocks. Each block consists of a multi-head self-attention layer followed by a feedforward MLP with GELU activations, residual connections, and layer normalisation. In our implementation, we use a single attention head.

The model receives entity and relation embeddings as input and processes them jointly through several Transformer blocks. The output is split into two parts: one passed through a softmax classifier over relations, and the other through a softmax classifier over entities.

While our architecture follows standard design choices, we emphasise that expressiveness is primarily introduced through the stacked non-linear transformations, particularly the use of Softmax layers over high-dimensional representations. This layered non-linearity enables richer modelling of interactions compared to simpler factorisation-based approaches.

## B.2. Optimisation

**Hyerparameters.** We fix the embedding dimension to 150 across all experiments. `ART` is trained using the AdamW optimizer (Loshchilov and Hutter, 2019) and the MLE objective, with an initial learning rate of 0.1. If the validation performance does not improve for five consecutive epochs, we apply learning rate decay with a multiplicative factor selected from $[0.1, 0.3, 0.5, 0.7, 0.9]$. We use a batch size of 1024 and apply dropout with a rate sampled uniformly in $[0, 0.5]$.

For the architecture, we perform a small-scale hyperparameter search over the number of Transformer blocks, using values from $[1, 2, 3, 4, 5]$, and the feedforward layer size, scaled as a multiple of the embedding dimension with multipliers from $[2, 3, 4, 5, 6, 7, 8]$. We fix the number of attention heads to 1 throughout. Positional encodings are omitted, since the input structure (head and relation embeddings) is fixed and semantically ordered.

We also explore different strategies for initialising and training the prior logits. Logits are either initialised uniformly or based on empirical training set frequencies, and are either kept fixed or treated as learnable parameters. Similarly, the Softmax Temperature is either fixed at 1.0 or learned during training.

**Model Selection** We explore several strategies for model selection, including monitoring validation loss, the average of training and validation losses, and Mean Reciprocal Rank (MRR). While validation loss is a standard choice, it may be unreliable on smaller knowledge graphs, where the validation set might not accurately reflect the true distribution. For consistency with prior work and fair comparison across models, we always select the checkpoint with the highest validation MRR. This also eliminates the need for a separate held-out set during information retrieval (IR) evaluation. We note, however, that selecting based on negative log-likelihood may yield better performance for generative tasks. The final model configurations and all settings required to reproduce the experiments are provided in the supplementary material.

## Appendix C. Additional Results

## C.1. Link Prediction

**Hits@$k$** Refer to Table 3 for Hits@$k$ results that complement the Link Prediction performance reported in the main text.

Table 3: **Hits@**$k$**.**

| Dataset | Model | H@1 | H@3 | H@5 | H@10 |
|---|---|---|---|---|---|
| FB15k-237 | ART | .249 | .378 | .442 | .530 |
| | NBF | .323 | .456 | .514 | .595 |
| | ComplEx | .247 | .369 | .433 | .520 |
| | ComplEx* | .247 | .369 | .433 | .520 |
| | ComplEx$^2$ | .217 | .331 | .389 | .469 |
| WN18-RR | ART | .424 | .461 | .480 | .503 |
| | NBF | .497 | .572 | .613 | .662 |
| | ComplEx | .433 | .485 | .511 | .545 |
| | ComplEx* | .433 | .485 | .511 | .545 |
| | ComplEx$^2$ | .342 | .423 | .448 | .471 |
| OGBL-BIO | ART | .769 | .877 | .913 | .946 |
| | NBF | .744 | .853 | .893 | .938 |
| | ComplEx | .760 | .879 | .916 | .950 |
| | ComplEx* | .760 | .879 | .916 | .950 |
| | ComplEx$^2$ | .774 | .888 | .923 | .954 |

## C.2. Information Retrieval

**Precision-Recall Curves**   These plots provide more detailed precision-recall curves for reference. The max-$F_1$ point used in the main text is indicated with a cross.

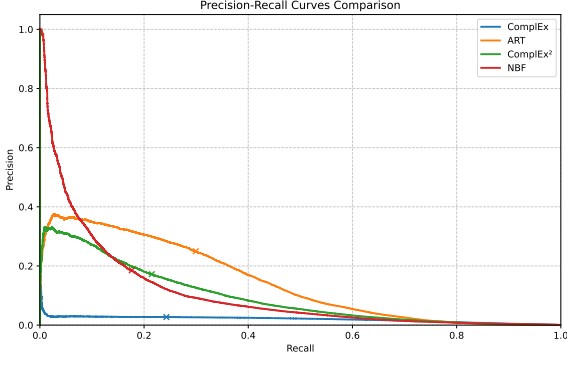

FB15k-237

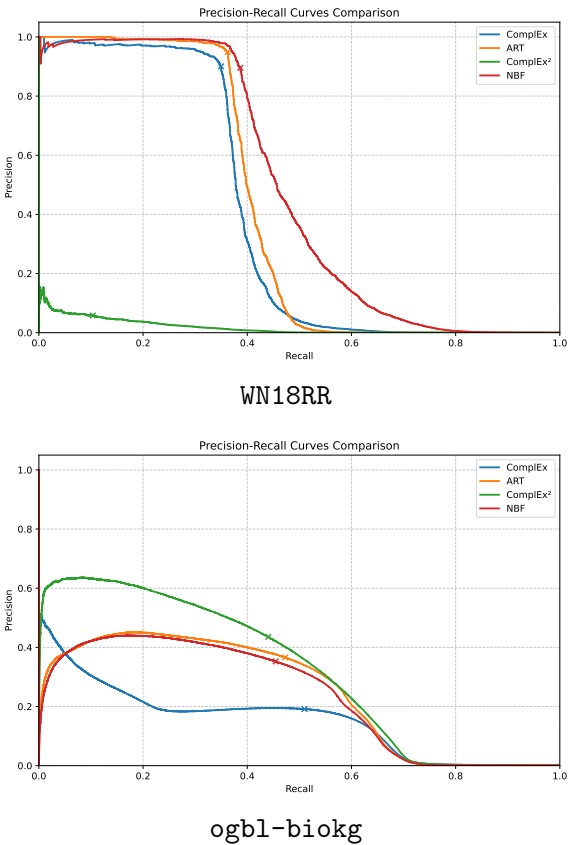

WN18RR

ogbl-biokg

**Optimal Threshold Analysis**  Post-Processing e.g. callibration does not improve Information Retrieval performance. Although callibration improves classification accuracy i.e. learning the optimal threshold, it does not improve the ranking performance, because the internal belief of the model can not be changed after training.

Table 4: **Optimal Threshold.**

| Dataset | Model | Threshold |
|---------|-------|-----------|
| FB15k-237 | ComplEx | 10.61 |
| | ComplEx* | 0.50 |
| WN18RR | ComplEx | 11.74 |
| | ComplEx* | 0.50 |
| OGBL-BIO | ComplEx | 11.21 |
| | ComplEx* | 0.50 |

## C.3. Generative Models for Knowledge Graphs

**Number of Parameters**  In traditional Tensor Factorization KGE models for discriminative tasks, increasing the embedding rank typically improves performance by enabling the model to better capture the structure of independent triples. However, as shown in

Table 5, this intuition does not hold for generative models that learn the joint distribution over knowledge graph triples.

Both `ART` and `ComplEx`$^2$ deviate from classical tensor factorization in both modeling approach and objective. Rather than scoring individual triples independently, generative models must capture complex dependencies across the graph. In this context, expressivity is more efficiently gained through a more complex, parameter-rich scoring function rather than by increasing the embedding dimension.

Table 5 highlights that `ART` allocates significantly more parameters to the scoring function, while `ComplEx`$^2$ relies entirely on larger embeddings. Despite having fewer total parameters, `ART` achieves greater expressive power through its design, underlining that parameterization of the scoring function plays a critical role in modeling joint distributions over triples.

Table 5: **Number Of Parameters.**

| Dataset | Model | Score | Emb | Total |
|---|---|---|---|---|
| FB15k-237 | ART | 15.71 | 2.25 | 17.96 |
| | ComplEx$^2$ | 0 | 29.56 | 29.56 |
| WN18RR | ART | 39.13 | 6.15 | 45.27 |
| | ComplEx$^2$ | 0 | 81.90 | 81.90 |
| OGBL-BIO | ART | 79.11 | 14.08 | 93.19 |
| | ComplEx$^2$ | 0 | 187.64 | 187.64 |

**Open/Closed World Assumption**  A knowledge graph embedding (KGE) model that assigns low probability to all unknown triples is of limited use under the Open World Assumption (OWA), where the goal is to discover new facts. In such settings, evaluation metrics that assume the Closed World Assumption (CWA)—like Expected Calibration Error (ECE)—become unreliable. They penalize the model for assigning high probability to triples that are simply unobserved, not necessarily false.

To highlight this issue, we focus on normalised models—`ART` and `ComplEx`$^2$—on the `OGBL-BIO` dataset. These models produce probabilities rather than scores, allowing us to construct meaningful precision-recall (PR) curves. We use the $F_1$-maximizing threshold from the main paper and evaluate performance under two contrasting assumptions: Optimistic (O), where all predictions above the threshold are assumed to be true, and Pessimistic (P), where they are assumed false.

While the main paper evaluates pessimism through $F_1$, we adopt a stronger form here using Mean Average Precision (MAP), which more harshly penalizes false positives. As shown in Table 6, the large gap between optimistic and pessimistic MAP values underscores how strongly the evaluation depends on unverifiable assumptions about the status of unknown triples.

This highlights a core problem: ECE and similar metrics assume a CWA-like interpretation where all unlabelled triples are considered negative. In reality, many such triples could be true, especially under OWA. When a model like `ART` assigns high probability to these, it may appear miscalibrated under CWA metrics—but may in fact be correct. These results

suggest that calibration-based evaluations under OWA should be interpreted with caution, and that metrics relying on CWA assumptions, like ECE or pessimistic MAP, can produce misleading comparisons.

Table 6: **ART assigns higher probability to more unknown triples at the $F_1$-optimal threshold**. Evaluation metrics like ECE assume a Closed World Assumption (CWA), treating all unlabelled triples as false—an assumption that can be misleading in Open World settings. To illustrate this ambiguity, we report MAP under both a Pessimistic (P) scenario, where all triples above the $F_1$ threshold are assumed false, and an Optimistic (O) scenario, where they are assumed true. Results on the OGBL-BIO dataset show substantial differences between the two, highlighting that CWA-based metrics can yield unreliable conclusions for normalised generative models.

| Model | Optimistic | Pessimistic |
|---|---|---|
| ART | **.780** | .245 |
| CompLEx$^2$ | .735 | **.318** |

