# OpenReview forum: "The ART of Link Prediction with KGEs"
_nesyconf.org/NeSy/2025/Conference_Phase_2 — NeSy 2025 - Phase 2 Poster_

### Official Review · Reviewer_LhKH · 2025-06-24
**Review of paper 16**

**Rating:** 4
**Confidence:** 3

**Review:**

This paper addresses a limitation in current Knowledge Graph Embedding models for Link Prediction, namely the lack of interpretability and comparability of scores across different queries. This issue affects the applicability of models in downstream tasks, as models can achieve high performance on standard query-based ranking metrics (like MRR) but assign inconsistent scores globally. The authors argue that this problem is often overlooked by standard evaluation protocols that focus on query-based ranking. Existing solutions are either based on post-processing methods (i.e., rescaling scores through a specific procedure) or probabilistic approaches which  aim to produce intrinsically normalized scores, meaning the scores can be interpreted as probabilities; examples of such approaches include autoregressive, energy-based and tensor factorization models.

The claimed contribution in this paper is twofold. First, a methodology for evaluating score normalization. Second, a novel knowledge graph embedding approach that guarantees normalization by design.

I found the paper potentially interesting, but also rather confusing and vague. In my opinion, the main weaknesses of the paper are as follows.

(1) The submission itself provides no clear explanation of its neurosymbolic AI relevance. The core contribution revolves around using an AutoRegressive Transformer to produce normalized KGE scores. While Transformers are neural, and KGEs are commonly used in conjunction with symbolic knowledge, the paper does not articulate how ART bridges the gap between neural and symbolic paradigms in a novel or significant neurosymbolic manner. Without such discussion, it is unclear how the proposed method aligns with the main topics of interest for a neurosymbolic learning and reasoning conference.

(2) Despite the model's "conceptual simplicity" claimed by the authors,  I found it impossible to fully understand how the proposed method works step by step with the detail provided in the paper, nor are the expected properties of the approach shown mathematically. For example, the paper claims that "Because each part of the factorised distribution is normalised this way, the full joint distribution is also normalised" which is a standard property of autoregressive models where factors are categorical and normalized. However, applying this to a complex, multi-layered Transformer architecture, which introduces "multiple non-linearities" requires a more thorough mathematical exposition to demonstrate that the overall function P(S,R,O) ndeed constitutes an exact joint distribution whose sum over all possible triples equals one.

(3) One of the paper's claimed contributions is "to establish a principled methodology to evaluate normalisation using a global ranking and IR metrics" However, I could not find a clear, comprehensive description of this methodology within the main body of the paper. While Section 3.1 briefly introduces a new metric and mentions constructing "a single global ranking across all queries" using standard IR metrics such as Precision, Recall, and max-F1,  it does not fully elaborate on the principled methodology in a way that feels like a distinct, formal contribution beyond stating the use of existing IR metrics on a globally ranked list. The description feels more like an application of existing metrics to a novel problem setting rather than a novel methodology itself.

(4) While the problem of unnormalized scores is acknowledged as challenging, the paper's explanation for its relevance could be more convincing for broader KGE utility beyond specific IR metrics.

(5) The model's efficient scoring for candidate generation is primarily in the forward direction (tail prediction). To support head queries (e.g., ?, r, o), the paper states that a known solution for this problem is simply to add inverse triples. This means ART's bidirectionality is achieved not intrinsically by its core architecture, but by data pre-processing and graph augmentation, which adds a layer of complexity and could be a consideration for very large KGs.

**Anonymity:**

Remain anonymous

---

### Official Review · Reviewer_QYxf · 2025-07-08
**Interesting techniques and useful results that only tangentially touch the main issues of neuro-symbolic integration.**

**Rating:** 6
**Confidence:** 3

**Review:**

The authors present an interesting discussion of link prediction in knowledge graphs, more precisely looking at knowledge graph embeddings that capture the main features in a graph. This is certainly relevant, and the paper has strong points as it looks into probabilistic layers to transformer-based models; the results are interesting but overall the contribution is not very clear from a neural-symbolic perspective. Basically it seems that we here use a neural network as a regression mechanism to capture probabilities (after some suitable transformations) so as to address a problem that uses symbols (because it is related to knowledge graphs); however, there is little mix between techniques that are intrinsically symbolic and techniques that are intrinsically neural. It seems that relevance to the meeting is somewhat hurt by this fact.

Overall, the text is easy to read without major problems. It would be nice if the authors could describe the neurally-based methods early and explain their role in the proposed methods.  One point: there are several problems with capitalization in the References (Bert, ai, bellman, pixelcnn, etc).

**Anonymity:**

Remain anonymous

---

### Official Review · Reviewer_okqb · 2025-07-08
**Interesting, Clear and Useful paper, introduces Auto-regressive Transformers for Link Prediction.**

**Rating:** 7
**Confidence:** 3

**Review:**

This well-written paper tackles a fundamental limitation in the evaluation of Knowledge Graph Embedding (KGE) models. The authors argue that existing KGE methods do not produce scores that are comparable across queries, making global confidence estimation problematic. To address this, they propose a new normalisation evaluation protocol. They also introduce the AutoRegressive Transformer (ART), a generative model normalised by design, and show empirically that ART achieves state-of-the-art performance.

- The problem is clearly defined and well motivated, exposing a gap in the existing literature.
- The writing is clear, and the proposed ART model is reproducible (code is provided along with usage instructions).
- The background section is well written and comprehensive.
- The experiments are supported by clear, easy-to-read tables (including supplementary material), and the choice of baselines is well justified.

However:

- The use of a Transformer architecture warrants the inclusion of wall-clock training times and memory usage measurements, to at least provide an idea of scalability across different-sized datasets.
- Some of the information on the Transformer architecture parameters found in Appendix B could be moved into the Architecture section (Section 3.2) for better clarity.

Overall, I found the paper both interesting and useful, and I recommend its acceptance.

**Anonymity:**

Remain anonymous